# Investigation of Glycosaminoglycans in Urine and Their Alteration in Patients with Juvenile Idiopathic Arthritis

**DOI:** 10.3390/biom13121737

**Published:** 2023-12-02

**Authors:** Elżbieta Lato-Kariakin, Kornelia Kuźnik-Trocha, Anna Gruenpeter, Katarzyna Komosińska-Vassev, Krystyna Olczyk, Katarzyna Winsz-Szczotka

**Affiliations:** 1Department of Clinical Chemistry and Laboratory Diagnostics, Faculty of Pharmaceutical Sciences in Sosnowiec, Medical University of Silesia, ul. Jedności 8, 41-200 Sosnowiec, Poland; elalato@op.pl (E.L.-K.); kkuznik@sum.edu.pl (K.K.-T.); kvassev@sum.edu.pl (K.K.-V.); olczyk@sum.edu.pl (K.O.); 2Department of Rheumatology, The John Paul II Pediatric Center in Sosnowiec, ul. G. Zapolskiej 3, 41-218 Sosnowiec, Poland; anna.gruenpeter@gmail.com

**Keywords:** juvenile idiopathic arthritis, urine glycosaminoglycans, chondroitin sulfate, dermatan sulfate, heparan sulfate, hyaluronic acid, biological therapy

## Abstract

(1) Background: In this study, we evaluated the modulation of urine glycosaminoglycans (GAGs), which resulted from etanercept (ETA) therapy in patients with juvenile idiopathic arthritis (JIA) in whom methotrexate therapy failed to improve their clinical condition. (2) Methods: The sulfated GAGs (sGAGs, by complexation with blue 1,9-dimethylmethylene), including chondroitin–dermatan sulfate (CS/DS) and heparan sulfate (HS), as well as non-sulfated hyaluronic acid (HA, using the immunoenzymatic method), were determined in the blood of 89 children, i.e., 30 healthy children and 59 patients with JIA both before and during two years of ETA treatment. (3) Results: We confirmed the remodeling of the urinary glycan profile of JIA patients. The decrease in the excretion of sGAGs (*p* < 0.05), resulting from a decrease in the concentration of the dominant fraction in the urine, i.e., CS/DS (*p* < 0.05), not compensated by an increase in the concentration of HS (*p* < 0.000005) and HA (*p* < 0.0005) in the urine of patients with the active disease, was found. The applied biological therapy, leading to clinical improvement in patients, at the same time, did not contribute to normalization of the concentration of sGAGs (*p* < 0.01) in the urine of patients, as well as CS/DS (*p* < 0.05) in the urine of sick girls, while it promoted equalization of HS and HA concentrations. These results indicate an inhibition of the destruction of connective tissue structures but do not indicate their complete regeneration. (4) Conclusions: The metabolisms of glycans during JIA, reflected in their urine profile, depend on the patient’s sex and the severity of the inflammatory process. The remodeling pattern of urinary glycans observed in patients with JIA indicates the different roles of individual types of GAGs in the pathogenesis of osteoarticular disorders in sick children. Furthermore, the lack of normalization of urinary GAG levels in treated patients suggests the need for continued therapy and continuous monitoring of its effectiveness, which will contribute to the complete regeneration of the ECM components of the connective tissue and thus protect the patient against possible disability.

## 1. Introduction

The extracellular matrix (ECM) is a porous, three-dimensional network of interacting molecules, and it plays an essential role in the structural and functional integrity of the body’s tissues and organs. The ECM, also a reservoir of growth factors and other bioactive molecules, regulates cell behavior such as proliferation, adhesion, migration, differentiation, apoptosis, and polarization [1]. The aforementioned matrix functions are directly related to the biophysical properties of glycosaminoglycan (GAGs) chains, found in the matrix mainly as covalently bound and less frequently non-covalently bound to proteoglycan core proteins (PGs) [2,3,4]. It is known that GAGs, due to their high density of negative electrical charge, enable proteoglycans to sequester water and divalent cations, as well as provide them with biomechanical functions, including the ability to fill intercellular spaces or “lubricate” surfaces. Moreover, GAGs play a key role in cell signaling, which modulates the aforementioned biochemical processes and maintains pro/anti-coagulant, pro/anti-inflammatory, or pro/anti-oxidative balance [3,4,5,6,7].

GAGs, as components of the dynamic ECM structure, undergo age-dependent remodeling, in which case modifications in the efficiency of GAGs synthesis or degradation may be part of a chain of pathogenetic changes leading to disease manifestations, e.g., juvenile idiopathic arthritis (JIA) [4,7,8]. Increased inflammatory process, observed in the course of JIA, which develops as a result of excessive activity of pro-inflammatory cytokines, including tumor necrosis factor-α (TNF-α), interleukin-1 (IL-1), IL-6, as well as reactive oxygen species (ROS) may contribute to proteolysis of core proteins of PGs, as well as the partial depolymerization of GAGs chains. The products of these catabolic processes, i.e., free glycan chains or their combination with fragments of proteoglycan cores, appear in the circulatory system and are transported to the liver or after a glomerular filtration pass to the urine [7,9]. Hence, the profile of GAGs in urine is an expression of tissue, i.e., cellular and extracellular metabolic turnover of PGs/GAGs [4,7,8,10]. Although in our previous studies [8] on children with JIA, we showed increased proteolytic–oxidative activity stimulated by the influence of pro-inflammatory cytokines, including TNF-α, which promotes the degradation of PGs, little is known about the effect on these transformations of the antagonist of the mentioned factor, i.e., etanercept (ETA, a recombinant TNF receptor-Fc fusion protein) used in the therapy of JIA. Based on the results of radiological evaluation [11], an improvement in musculoskeletal performance was demonstrated due to the inhibition of the progression of degenerative joint changes in ETA-treated patients; however, the above data were not confirmed by the results of biochemical tests performed in readily—non-invasive—available, biological material.

Hence, in this study, we evaluated the modulation of urine-sulfated glycosaminoglycans (sGAGs), including chondroitin–dermatan (CS/DS), heparan sulfate (HS), and non-sulfated GAG, i.e., hyaluronic acid (HA) that resulted from ETA therapy in patients in whom methotrexate therapy failed to improve their clinical condition. As far as we know, the present study is the first attempt to evaluate changes in the concentration of both total GAGs and their individual types in the urine of JIA patients before and during biological therapy. In addition, the present study is a continuation of our previous studies, in which we evaluated serum/plasma concentrations of selected ECM components that can be used as potential biomarkers of articular cartilage status in pediatric patients [4,8,12,13,14].

## 2. Materials and Methods

### 2.1. Patients and Samples

A total of 59 Caucasian children of both sexes, including 40 girls and 19 boys, aged from 2 to 12 years, participated in the study. Pediatric rheumatologists from the Rheumatology Outpatient Clinic of the John Paul II Independent Public Health Care Center for Pediatrics in Sosnowiec diagnosed them with JIA. All the patients were diagnosed and classified as oligoarthritis or polyarthritis, according to the International League of Rheumatological Societies criteria and using the exclusion list [15,16,17]. The clinical diagnosis was confirmed by the results of diagnostic laboratory tests, summarized in Table 1. The Juvenile Arthritis Disease Activity Score-27 (JADAS-27) was used to assess activity in all patients. Patients were treated with oral glucocorticoids (Encorton, EC, at a maximum dose of 1 mg of prednisone equivalent per kilogram per day, with gradual dose reduction), sulfasalazine (SSA, 30 mg per square meter of body surface area), and methotrexate (MTX, ≤15 mg per square meter of body surface area, once a week). The ACR Pediatric criteria were used to assess response to JIA treatment [17]. The group of patients with newly diagnosed and previously untreated JIA was defined as TP, while the group of JIA patients who achieved remission with MTX, EC, and SSA (10.80 ± 0.35 months after the beginning of the therapy) was defined as TR. 

Patients in whom the above drugs did not improve their clinical condition received ETA therapy (according to the Polish Therapeutic Programme using TNF blockers, i.e., B.33) at a dose of 0.4 mg/kg body weight (up to a maximum dose of 25 mg) via subcutaneous injection twice a week, 3–4 days apart. After 3 months of biological therapy, both EC and SSA were withdrawn. In these JIA patients, the assessment of urine glycosaminoglycans alterations was performed both before the initiation of biological therapy (T0) and in the same patients at the following intervals, i.e., after the third (T3), the sixth (T6), the twelfth (T12), the eighteenth (T18), and the twenty-fourth (T24) month of ETA treatment, i.e., after clinical improvement.

Patients with traumatic injuries or musculoskeletal surgeries performed in the last 3 years, patients with autoimmune and metabolic diseases including diabetes mellitus, cancer, kidney disease, liver disease, and chronic infections, and patients with mental illness or other conditions that impede physician–patient cooperation were excluded from this study. 

The reference material for the study consisted of urine samples obtained from 30 healthy children (20 girls and 10 boys) corresponding to the age of patients with JIA during routine control tests at the Medical Diagnostic Laboratory of ZPOZ in Jedrzejow. Children who qualified for the control group did not suffer from a disease requiring hospitalization during the previous year. Moreover, they were not treated pharmacologically just before the studies, and the results of their routine laboratory tests, including the evaluation of hematological and biochemical parameters and indicators of inflammation, did not deviate from the values accepted as normal for this age group [18].

The characteristics of laboratory parameters of blood and urine samples of healthy children and patients with JIA are shown in Table 1.

The project was approved by the Local Bioethics Committee (No. KNW/0022/KB/168/18) of the Medical University of Silesia in Katowice. Informed consent was obtained from healthy volunteers and patients with JIA and/or their parents or guardians

### 2.2. Preliminary Development of Biological Material

Urine from healthy children and patients with JIA who qualified for the study was collected in a manner ensuring its optimal diagnostic value. Thus, the urine was obtained from the first-morning micturition from the middle stream and collected after thoroughly washing the urethral outlet. Urine was collected at the appropriate time, as shown in Figure 1. The biological material was placed into disposable sterile containers. Prior to assays, the urine was centrifuged at 15,000× *g* for 10 min at 4 °C. The centrifuged urine samples obtained from both healthy individuals and JIA patients were divided into portions and stored at −80 °C before the initiation of the study. Each matrix parameter was determined in one day. Hence, the inter-assay variation was insignificant.

### 2.3. The Assay of the Concentration of Urine-Sulfated Glycosaminoglycans

The total pool of urinary GAGs consists of sulfated glycosaminoglycans (sGAGs), including CS/DS, HS, and non-sulfated HA. It should be noted that keratan sulfates (KS) were not included in the pool of urinary sulfated glycans, as these macromolecules are present in biological fluids only in trace amounts due to the negligible content of these glycans in tissues.

Determination of the total concentration of sGAGs and their individual fractions, i.e., CS/DS and HS in urine samples, was carried out using the Blyscan Sulfated Glycosaminoglycan Assay diagnostic test from Biocolor Ltd., Carrickfergus, UK. Determination of the aforementioned compounds consisted of their complexation reaction (both before and after application of the HS degrading agent) with the staining reagent 1,9-dimethylmethylene blue (DMB), according to the procedure described previously [4]. The within-run repeatability for the sulfated glycosaminoglycan assays was less than 6%.

### 2.4. The Assay of the Concentration of Urine Hyaluronic Acid

Determination of hyaluronic acid concentration in the studied urine samples was performed using an enzyme-linked immunosorbent assay (ELISA), with a minimum detection of 13.3 ng/mL using the Hyaluronic Acid Plus diagnostic test from TECO Medical Group.

The analytical technique used involved the reaction between the hyaluronic acid molecule to be assessed, and the hyaluronic acid binding protein (HABP) immobilized on the surface of the solid phase, i.e., the reaction well. The assay procedure is described in our previous publication [4]. The within-run repeatability for the HA assay was less than 5%.

Because of the probable presence of a circadian rhythm of urinary excretion of glycosaminoglycans and because of the risk of pre-laboratory errors associated with obtaining a morning urine sample from children, the obtained values of urinary GAGs concentrations expressed in mg were converted to g creatinine in this study.

### 2.5. The Assay of the Concentration of Urine Creatinine

The creatinine concentration in the urine samples was determined using an enzymatic method using the CREATININE assay from ELITech Group Clinical Systems, using a Selectra Pro-M automated biochemical analyzer. The within-run repeatability for creatinine determinations was less than 1.9%.

### 2.6. Statistical Analysis 

The results obtained were subjected to statistical analysis using the software STATISTICA (version 13.3 by TIBCO StatSoft Poland). The analysis included: checking the normality of the distribution for a given group, using the Shapiro–Wilk test; checking the equality of variance, using the Levene’s test; descriptive characteristics for traits with a normal distribution in the form of the arithmetic mean and the standard deviation; testing the significance of differences in mean values of a given trait for the control group and the study group before and after treatment for traits with a normal distribution by applying the Student’s *t*-test for independent samples; testing the significance of differences in mean values of a given trait for individuals from individual study groups for traits with a normal distribution by applying an analysis of variance (ANOVA) with repeated measures, while Tukey’s multiple comparisons test was used as a post hoc test; descriptive characteristics for traits with a non-normal distribution, in the form of the median and the interquartile range lower quartile and upper quartile; testing the significance of differences in mean values of a given trait for the control group and the study groups for traits with an asymmetric distribution by applying the Mann–Whitney U test and for both study groups for traits with an asymmetric distribution using Wilcoxon’s paired rank test; and assessing the relationship between two variables using Pearson’s r correlation coefficient. A significance level of *p* < 0.05 was assumed for each statistical analysis.

## 3. Results

The results of the determination of total sGAGs as well as CS/DS, HS, and HA in the urine of subjects of all study groups are shown in Table 2. They are the results of the group of healthy children (C), the group of children with newly diagnosed and previously untreated JIA (TP), the group of afflicted children who achieved remission with MTX, SSA, and EC (TR), and those with aggressive disease before the start of ETA treatment (T0), and after twenty-four (T24) months of biological treatment, adjusted for gender.

### 3.1. The Urine Levels of sGAGs, CS/DS, HS, and HA in Healthy Children and JIA Patients

There was a significant (*p* < 0.05) reduction in urinary sGAGs in patients with newly diagnosed JIA compared to urinary glycans in the control children (C). It was also shown that therapy with MTX, SSA, and EC contributed to clinical improvement in some patients (TR) but did not normalize the urinary sGAG concentration in these patients. These concentrations were still significantly (*p* < 0.05) lower compared to the controls. Glycan concentrations were also significantly (*p* < 0.05) lower in patients in whom the aforementioned therapy did not result in clinical improvement (T0). Similarly, urinary sGAG concentrations in T24 children were still significantly (*p* < 0.05) lower compared to urinary concentrations in healthy children. A similar nature of sGAG changes characterized the affected girls. In boys, significantly (*p* < 0.05) lower urine sGAG concentrations were found only in the TR group compared to the controls.

As can be concluded from the data presented in Table 2, the urinary concentrations of CS/DS in patients in the TP and T0 groups were significantly (*p* < 0.05) lower compared to the controls. In contrast, the urinary concentrations of these glycans in patients with a compensated clinical condition were comparable (*p* > 0.05) to those in the urine of healthy children. On the other hand, urinary concentrations of CS/DS in the urine of afflicted girls from all study groups, i.e., TP, TR, T0, and T24, were significantly (*p* < 0.05) lower compared to the controls. For the afflicted boys, significantly (*p* < 0.05) lower concentrations of the described glycans were shown only in the TP group.

Compared to sGAGs and CS/DS, the nature of the changes for HS and HA was shown to be different. Concentrations of both glycans increased significantly in the urine of the patients with active disease. Thus, relative to the controls, HS concentrations were significantly (*p* < 0.05) higher in the following groups: TP and T0, whereas HA was significantly (*p* < 0.05) higher in the TP and T0 groups. The urinary excretion of the glycans in question in the TR and T24 group patients was comparable (*p* > 0.05) to that in healthy children. Furthermore, in the course of JIA, the changes in HS and HA concentrations in the urine of afflicted girls are characterized by similar trends to those of all patients without sex classification. In contrast, in afflicted boys, HS concentrations were significantly (*p* < 0.05) higher compared to the controls only in the TP group, while HA concentrations were higher in the T0 group.

### 3.2. Dynamics of Changes in the Concentration of sGAGs, CS/DS, HS, and HA in the Urine of JIA Patients

Figure 2a–d, respectively, illustrate the dynamics of changes in the concentration of sGAGs, CS/DS, HS, and HA in the urine of children with JIA both before (T0) and after 3 (T3), 6 (T6), 12 (T12), 18 (T18), and 24 (T24) months of applying ETA, accepting as a starting value the concentration of glycans shown in the urine of the afflicted children before the first dose of the TNF-α inhibitor (T0).

Six months of ETA therapy was shown to contribute to a decrease in urinary sGAG levels in treated children with JIA, while continued therapy contributed to an upward trend. The level of urine sGAGs in patients with active JIA who qualified for biological treatment (T0) was significantly higher compared to the concentration in the urine of the patients in subsequent months of treatment, i.e., T3 (*p* < 0.05) and T6 (*p* < 0.005). It was also shown that the concentration of the macromolecules in question in the urine of patients after twelve (T12, *p* < 0.05), eighteen (T18, *p* < 0.05), and twenty-four months (T24, *p* < 0.005) of ETA use was statistically higher compared to the concentration found in patients after six months (T24) of anti-cytokine therapy (Figure 2a).

A similar nature of the change to that of sGAGs was shown for CS/DS, whose concentrations were lowest at T6 (Figure 2b). Accordingly, significantly higher glycan concentrations characterized patients in groups T18 (*p* < 0.05) and T24 (*p* < 0.01) compared to urinary CS/DS concentrations of patients in the T6 group. Moreover, the concentration of the assessed compounds in the T24 group was significantly (*p* < 0.05) higher than the CS/DS concentration in the T3 group.

Contrary to sGAGs and CS/DS, there was a gradual decrease in the amount of urinary excretion of HS with the duration of ETA therapy (Figure 2c). It was shown that the urinary HS concentration of patients in the T12 group was significantly (*p* < 0.05) lower than the T0 and T3 groups. Furthermore, significantly (*p* < 0.05) lower HS concentrations were shown in both the group of patients treated with biologics for eighteen months (T18) and for twenty-four months (T24) in comparison to those shown in the T0, T3, and T6 groups.

The use of biological therapy in patients with JIA also led to a significant decrease in urinary HA excretion, which was observed during treatment (Figure 2d). The level of urine HA of the patients with active JIA who qualified for biological treatment (T0) was significantly (*p* < 0.0001) higher compared to the concentration in the urine of the patients in subsequent months of treatment, i.e., T6, T12, T18, and T24. In addition, the concentration of assessed glycan in the urine of the patients in the T18 and T24 groups was statistically significantly (*p* < 0.05 and *p* < 0.005, respectively) lower compared to the concentration of assessed molecules in the urine of children in the third month (T3) of biological therapy.

### 3.3. Correlation Analysis between Urine sGAGs, CS/DS, HS, HA, and Serum CRP Levels in JIA Patients

In order to achieve the primary aim of the study, correlations between the urinary concentrations of sGAGs, CS/DS, HS, and HA in all study groups, i.e., C, TP, TR, T0, and T24, and the values of an indicator of the inflammatory process, i.e., the C-reactive protein in the blood of these patients, were assessed (Table 3).

Thus, the statistical analysis of the obtained test results revealed the presence of significant associations between the amount of urinary excretion of sGAGs and the value of serum CRP concentration in healthy children [(C), (r = 0.5108; *p* = 0.0039)], in the patients with remission of JIA vs. those being treated with MTX, SSA, and EC [(TR), (r = 0.7742; *p* = 0.00001)], and in the patients after two years of ETA therapy [(T24), (r = 0.7377; *p* = 0.0000)].

Correlations were also revealed between urinary CS/DS and serum CRP levels in both healthy children from the control group (r = 0.6417; *p* = 0.0001) and in the patients with JIA from the following groups: TP (r = −0.5821; *p* = 0.00000), TR (r = 0.7821; *p* = 0.00001), T0 (r = −0.6508; *p* = 0.00002), and T24 (r = 0.7611; *p* = 0.00000).

In addition, a significant association was found between HS and CRP in children with newly diagnosed, previously untreated JIA [(TP), (r = 0.7221; *p* = 0.0000)], the patients with remission of JIA resulting from treatment with MTX, SSA and EC [(TR), (r = 0.7104; *p* = 0.0001)], and the patients with aggressive disease before ETA treatment [(T0), (r = 0.8048; *p* = 0.00000)]. 

Significant associations were found between HA and CRP in patients from the following groups: TP (r = 0.6902; *p* = 0.00000), TR (r = 0.6389; *p* = 0.0008), T0 (r = 0.6360; *p* = 0.00004), and T24 (r = 0.5186; *p* = 0.0014).

## 4. Discussion

The hypothesis that ECM structure abnormalities already occur at the early stages of JIA is supported by the changes in urinary sGAG excretion demonstrated in this study in patients with newly diagnosed, previously untreated arthropathy. Significantly lower urinary concentrations of sGAGs in JIA patients, especially in the afflicted girls, compared with healthy children, did not normalize with disease-modifying treatment with MTX, SSA, and EC, as well as with anti-cytokine treatment. The results obtained are consistent with the results of the research of other authors [4,19]. It should be pointed out, however, that Shevchenko et al. [19] evaluated glycan levels by measuring daily urinary excretion of uronic acids in patients with different durations of JIA. The researchers observed lower urinary concentrations of the tested markers, especially in patients with a disease duration of 13 to 24 months. Moreover, contrary to our results, in the female patients, the authors showed higher concentrations of uronic acids compared to the results obtained in the male group and the whole group of healthy subjects, without taking their gender into account. Females, according to the cited researchers, are predisposed to the development of osteoarthritis secondary to JIA [19].

Although we found significantly lower concentrations of the assessed compounds in afflicted girls, we can confirm this thesis. It is known that autoimmune diseases, including JIA, involving all organs and systems develop in approximately 8% of the population, and 78% affect females. This predisposition of the females is likely related to the stimulating effects of estrogens and prolactin on the humoral immune response and, in addition, to the pro-inflammatory actions of these hormones. On the contrary, androgens exert an inhibitory effect on both humoral and cellular immune responses, thus appearing to represent a group of natural anti-inflammatory hormones [20]. 

Given that the values of urinary GAG concentrations reflect the profile of GAGs in blood, it is important to point out the similarity of the results obtained in the present study with those in the studies assessing GAG concentrations in blood as well as joint fluid [4,12,21] of people with JIA. It has been demonstrated that the concentration of total GAGs in the aforementioned fluids of JIA patients does not normalize due to the disease-modifying treatment with MTX [4,12,21]. The changes in GAG concentrations demonstrated by the cited authors were mainly due to the changes in CS concentrations, which are the quantitatively dominant fraction of glycans circulating in the blood [4,21].

The results obtained in the present study assessing the excretion of CS/DS in the urine of the patients with JIA correspond with the results of the previously described studies [4]. Indeed, we confirmed that CS/DS, despite the significant reduction in the concentration of these glycans found in patients’ urine, constitute the predominant fraction of urine sGAGs. These glycans appear to directly determine the profile of changes in the total pool of urine sGAGs in untreated patients and those with the aggressive form of JIA, i.e., disease requiring anti-cytokine therapy. The low concentrations of CS/DS in the urine of the patients with untreated JIA appear to be the result of increased tissue degradation of these compounds already occurring in the early, still preclinical stages of the disease. At the onset of clinical symptoms of the disease, the pool of the tissue sGAGs is significantly reduced, and the processes of their synthesis do not balance the magnitude of their degradation. Therefore, the concentration of sGAGs in the urine of patients from the analyzed groups and the CS/DS concentration in the patients with active forms of the disease are significantly lower compared to the controls. Similar conclusions were reached by Maldonado and Nam [22], who indicated that aggrecan and collagen degradation was an important event in the early stages of osteoarthritis. In our previous studies, we described in detail the mechanisms of proteolytic and oxidative degradation of ECM components and the pathways leading to changes in their synthesis in patients with JIA [4,8,11,12].

The significantly lower concentration of CS/DS in the urine of children with arthropathy, which correlates negatively with CRP values, probably reflects a “depletion” of the tissue pool of these GAGs. The above is also due to their excessive consumption resulting from the antioxidant, antiproteolytic, and anti-inflammatory functions of CS [9,23,24,25]. Indeed, these biomolecules are able to “scavenge” hydroxyl radicals and superoxide anion radicals, limit cellular damage caused by a MMPs/TIMPs imbalance, and regulate NO and prostaglandin E2 synthesis [23,24,25,26]. Moreover, these glycans inhibit the expression and activity of pro-inflammatory enzymes, including cyclooxygenase 2 or phospholipase A2, and reduce the synthesis of pro-inflammatory cytokines, including TNF-α or IL-1β [26]. As a result, the degradation of ECM components is inhibited, and their synthesis is restored.

A different nature of the changes in the urine of children with JIA, in relation to sGAGs and CS/DS, was found in the case of HS and HA. Significantly higher urinary concentrations of these glycans were found in children with untreated JIA and those with aggressive arthropathy compared with the controls. Other studies in children with arthropathy have found no significant changes in the excretion of the GAGs in question [4]. A different nature of the alterations of HS in relation to the other sulfated glycans shown in the present study is probably due to differences in the position of these compounds in cartilage, as well as their interaction with receptors that may influence the development of arthropathy, as described by Shamdani et al. [27]. Moreover, an increased expression of some of the HS core proteins in osteoarthritis, including perlecan, syndecan 1, or syndecan 4 has been shown [28]. The significant increase in urinary HS concentrations observed in children with active forms of JIA, strongly associated with the severity of the inflammatory process, appears to be one of the factors promoting the development of arthropathy. The stimulatory effects of HS on the biosynthesis of catabolic factors, including MMP-3, MMP-13, and ADAMTS-4, and the inhibitory effects on the mRNA levels of anabolic markers, including COL2, ACAN, SOX9, and VEGF, in mouse articular chondrocytes are known [27]. Although the above influences have not been confirmed in patients with JIA, we cannot exclude such mechanisms.

Similar to HS, we found a significant increase in urinary HA excretion in subjects with active forms of JIA, especially in female patients, which was related to the severity of inflammation. In contrast to the results of the present study, a reduction in urinary HA excretion was demonstrated in patients with newly diagnosed arthropathy, in whom the administered treatment was effective and led to remission of the condition in all subjects [4]. However, the molecular weight of these hyaluronan chains is not known, but it is very likely to be low. This value seems to prove that the hyaluronan chains present in the urine of afflicted children are short and are products of increased depolymerization of the discussed glycan. This thesis is supported by the demonstrated reduction in the molecular weight of HA in synovial fluid in patients with knee osteoarthritis [29]. It should be noted that in synovial fluid, HA is the main component synthesized by synoviocytes, fibroblasts, and chondrocytes, and the molecular weight of its native molecule, as the so-called high molecular weight HA (HMM), ranges from 2 × 10^5^ to 2 × 10^6^ Da. On the other hand, low molecular weight HA (LMM), defined as a compound with a molecular weight lower than 3 × 10^5^ Da, is formed via the degradation of HMM-HA or is synthesized de novo in processes catalyzed by hyaluronan synthases during inflammation [30,31,32].

Changing the balance of HMM-HA and LMM-HA in favor of LMM-HA may lead to chronic inflammation and, consequently, to increased expression of pro-inflammatory mediators through abnormal receptor signaling. It is known that HA molecules of different molecular weights have separate molecular and cellular mechanisms and various, often opposing, biological effects, which result from their interaction with different classes of cell surface receptors [30,31,32]. Among the HA receptors, the key role in inflammatory diseases is played by CD44, which interacts with many other ligands besides HA, including collagen, fibronectin, laminin, chondroitin sulfate, or L and E selectin, and thereby regulates the ECM metabolism [33]. CD44-mediated signaling affects both chondrocyte survival and apoptotic (chondroapoptotic) pathways [34]. For example, in a CD44-dependent mechanism, HA fragments formed in the processes of free radical depolymerization may reciprocally increase the synthesis of nitric oxide [34]. The above is one of the mechanisms through which LMM-HA stimulates inflammatory processes, resulting in the reduced viscosity of synovial fluid with subsequent mechanical damage to the cartilage. In light of the above data, it seems that the glycosaminoglycan profile observed in the urine of children with JIA, which is different for individual fractions of these macromolecules, reflects their different sensitivity to the influence of factors leading to the development of the disease, as well as the different participation of glycans in etiopathogenetic mechanisms of arthropathy. Due to the multifunctionality of GAGs, the above-described CS/DS transformations seem to prevent the development of arthropathy, while the HS and HA transformations favor its manifestation.

The above thesis seems to be confirmed by our next results, which revealed opposite trends in changes in CS/DS concentrations in relation to those characterizing HS and HA in the children with aggressive JIA, both before and during anticytokine treatment. Thus, while the amount of CS/DS excretion in the urine of biologically treated patients gradually increased with the duration of therapy, in the case of HS and HA, this amount decreased. Therefore, the less pronounced increase in the excretion of total sGAGs in the urine of biologically treated children, compared to CS/DS, is an expression of compensation caused by a decrease in the concentration of HS and HA in these patients. What is more, different relationships between the above-mentioned types of glycans and the concentration of CRP in patients’ blood were shown. In patients who qualified for etanercept treatment (T0 group), low urinary CS/DS concentrations were observed in those with high CRP values, confirmed using the negative Pearson’s r correlation coefficient. On the other hand, positive values of the mentioned coefficient were characterized by the relationships between CRP, HS, and HA. Similar positive associations were shown in patients after 2 years of treatment with a TNF-α inhibitor (T24 group) for sGAGs, CS/DS, and HA.

Due to the key importance of TNF-α in “organizing” the inflammatory immune response in patients, resulting, among others, from stimulating the release of matrix catabolic factors, including MMPs, cathepsins, or ROS, therapies that neutralize the action of this factor are highly effective in the treatment of chronic inflammatory and autoimmune pathologies, including JIA [35]. The transformation of HA in JIA patients seems to be directly related to the influence of TNF-α. The team of Lim et al. [36] proved that the described factor and IL-1β increased the expression of HAS-1 and HAS-2, i.e., enzymes involved in hyaluronan biosynthesis, in fibroblast cultures. However, it is not known whether TNF-α-stimulated HA synthesis leads to the formation of HMM-HA or pro-inflammatory LMM-HA chains.

Therefore, according to the above mechanism, the consequence of using an inhibitor of the factor in question, i.e., ETA, in patients with an aggressive form of the disease, a decrease in the pool of newly synthesized HA molecules will be observed, and consequently, a decrease in the concentration of this glycan in the urine of treated patients with arthropathy.

In addition to improving the clinical condition, inhibiting the multidirectional effects of TNF-α due to the use of ETA in patients with JIA should lead to the regeneration of ECM components. However, the lack of normalization of sGAG concentration in the urine of biologically treated patients, demonstrated in this study, seems to indicate the presence of subclinical inflammation in these patients, which is probably related to the autoimmunity process. Autoantibodies directed against glycosaminoglycans are found in the blood of patients with inflammatory joint diseases [37].

## 5. Conclusions

The metabolisms of glycans during JIA reflected in their urine profile, depend on the patients’ sex and the severity of the inflammatory process. The remodeling pattern of urinary glycans observed in patients with JIA indicates the different roles of individual types of GAGs in the pathogenesis of osteoarticular disorders in sick children. Furthermore, the lack of normalization of urinary GAG levels in treated patients suggests the need for continued therapy and continuous monitoring of its effectiveness, which will contribute to the complete regeneration of the ECM components of the connective tissue and thus protect the patient against possible disability. In children with JIA, frequently irreversible structural and functional changes in the musculoskeletal system may occur as a result of an impaired ECM metabolism. However, a large study sample is needed to assess the usefulness of urinary GAG determinations as biomarkers of JIA progression. The relatively small sample size in this study, resulting from the low prevalence of JIA among children in Poland (5–6.5 per 100,000 children) [38], constitutes the main limitation of this work.

## Figures and Tables

**Figure 1 biomolecules-13-01737-f001:**
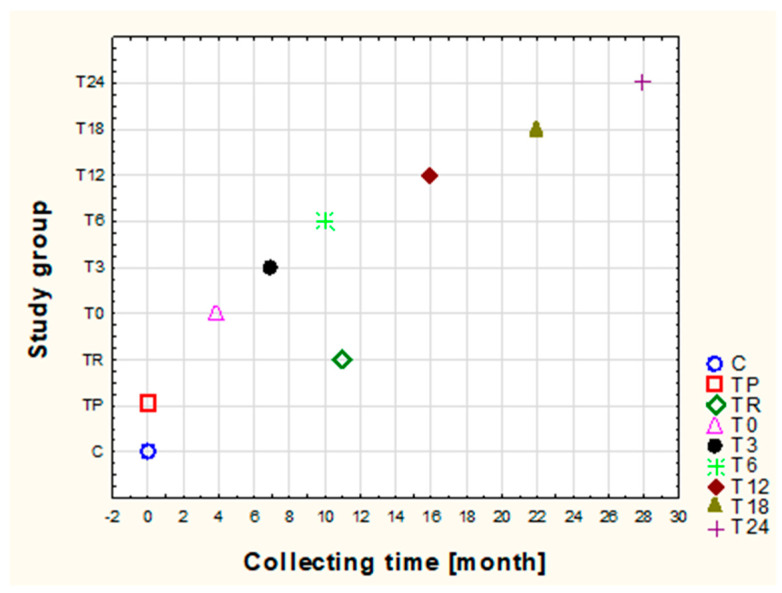
Timeline showing when the urine samples from the different groups were collected (C, control subjects; (TP), patients with newly diagnosed and previously untreated JIA; (TR) JIA patients who achieved remission with MTX, EC, and SSA; children with JIA both before (T0) and after 3 (T3), 6 (T6), 12 (T12), 18 (T18), and 24 (T24) months of applying ETA).

**Figure 2 biomolecules-13-01737-f002:**
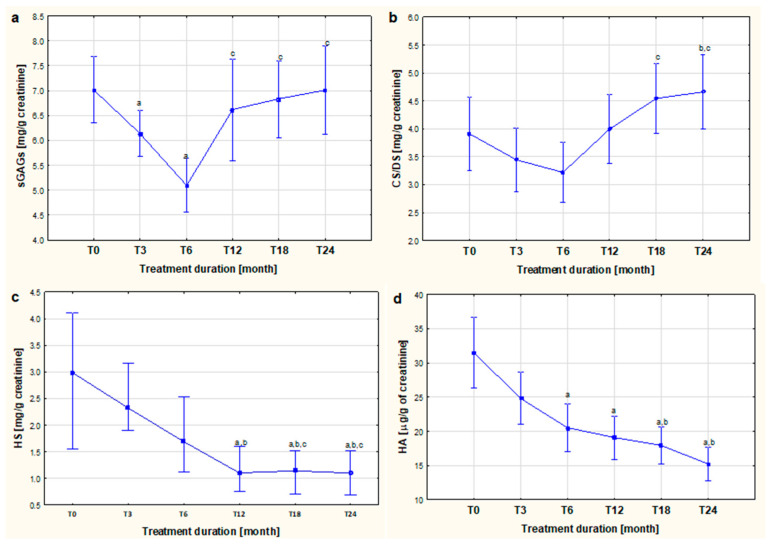
The dynamics of changes in the concentration of sGAGs (**a**), CS/DS (**b**), HS (**c**), and HA (**d**) in the urine of children with JIA both before (T0) and after 3 (T3), 6 (T6), 12 (T12), 18 (T18), and 24 (T24) months of applying ETA. ^a^ *p* < 0.05 compared to T0, ^b^ *p* < 0.05 compared to T3, ^c^ *p* < 0.05 compared to T6.

**Table 1 biomolecules-13-01737-t001:** The clinical data of control subjects and JIA patients.

Parameter	Control Subjects *n* = 30			JIA Patients	JIA Patients				
		TP *n* = 59	TR *n* = 24	T0 *n* = 35	During ETA Treatment				
					T3 *n* = 35	T6 *n* = 35	T12 *n* = 35	T18 *n* = 35	T24 *n* = 35
**Age (years)**		6.46 ± 2.78	7.53 ± 2.49	6.90 ± 2.22	7.14 ± 2.30	7.37 ± 2.16	7.98 ± 2.31	8.65 ± 2.55	9.01 ± 2.54
**Sex (F/M)**	20/10	40/19	17/7	23/12	23/12	23/12	23/12	23/12	23/12
**JADAS-27**	-	35.00 (25.10–40.00)	0.25 (0.00–5.00)	41.50 (36.50–49.50)	17.50 (15.50–21.50)	9.50 (8.00–13.50)	2.50 (1.00–4.00)	1.00 (1.00–1.50)	0.50 (0.00–1.00) ^a^
**Drugs**	-	MTX, EC, SSD	MTX, SSD	MTX, EC, SSD	ETA, MTX, EC, SSD	ETA, MTX	ETA, MTX	ETA, MTX	ETA, MTX
					**Blood**				
**WBC [10^3^/µL]**	6.9 ± 2.06	10.04 ± 4.03 *	7.14 ± 2.31	18.72 ± 2.18 *	7.08 ± 1.64	6.36 ± 2.14	6.72 ± 1.74	6.45 ± 1.68	6.68 ± 2.06
**RBC [10^6^/µL]**	4.88 ± 0.29	4.48 ± 0.41	4.62 ± 0.36	3.89 ± 0.43 *	4.23 ± 0.61	4.54 ± 0.40	4.58 ± 0.38	4.66 ± 0.41	4.58 ± 0.52
**Hb [g/dL]**	13.56 ± 0.89	11.58 ± 1.38 *	12.97 ± 1.16	11.35 ± 1.27 *	12.47 ± 1.05	12.93 ± 1.45	13.3 ± 1.73	13.78 ± 1.45	13.54 ± 1.36
**Ht [%]**	41.02 ± 3.9	35.35 ± 3.61	37.17 ± 7.39	37.19 ± 3.57	39.73 ± 2.52	38.65 ± 4.47	39.65 ± 5.26	40.44 ± 5.07	39.31 ± 4.15
**PLT [10^3^/µL]**	291 ± 76.23	405.47 ± 129.1	351.20 ± 93.78	344.32 ± 70.15	362.21 ± 54.99	317.16 ± 65.61	328.4 ± 85.22	312.25± 75.34	336.92 ± 54.21
**Cr [mg/dL]**	0.68 ± 0.14	0.71 ± 0.21	0.73 ± 0.17	0.71 ± 0.48	0.77 ± 0.13	0.73 ± 0.14	0.79 ± 0.14	0.89 ± 0.10	1.26 ± 0.49 *
**GPT [U/L]**	17.56 ± 5.43	18.46 ± 6.78	16.91 ± 8.08	17.97 ± 9.48	18.71 ± 6.29	17.26 ± 10.09	21.75 ± 21.27	24.5 ± 7.21	22.57 ± 24.73 *
**GOT [U/L]**	16.28 ± 5.08	17.58 ± 5.23	19.97 ± 7.95	20.11 ± 5.54	21.11 ± 5.32	20.11 ± 7.83	22.7 ± 11.10	20.19 ± 7.85	21.19 ± 16.56 *
**ESR [mm/h]**	7.9 ± 4.01	42.0 ± 27.0 *	13.11 ± 7.21	42.95 ± 15.89 *	18.68 ± 12.09	9.68 ± 9.89	9.80 ± 11.51	9.69 ± 11.94	7.08 ± 19.88
**CRP [mg/L]**	0.69 (0.40–1.11)	20.25 (9.11± 40.00) *	2.76 (1.73–7.56)	23.83 (18.5–33.79) *	13.98 (11.69–6.12)	0.79 (0.38–5.16)	0.74 (0.32–2.6)	0.45 (0.2–1.2)	0.43 (0.23–1.61) *
**RF**	-	N	N	N	N	N	N	N	N
**Urine**									
**Cr [mg/dL]**	85.46 ± 54.89	95.55 ± 58.88	88.32 ± 81.25	96.03 ± 21.95	118.91 ± 91.36 *	139.95 ± 88.06 *	136.42 ± 91.67 *	142.39 ± 76.9 *	136.12 ± 101.2 *
**Protein [mg/dL]**	N	N	N	N	N	N	N	N	N

Results are expressed as mean ± SD or medians (quartile 1–quartile 3); ^a^ *p* < 0.05 compared to untreated JIA patients; * *p* < 0.05 compared to the control group; F/M, female/male; ETA, etanercept; (TP), patients with newly diagnosed and previously untreated JIA; (TR) JIA patients who achieved remission with MTX, EC, and SSA; children with JIA both before (T0) and after 3 (T3), 6 (T6), 12 (T12), 18 (T18), and 24 (T24) months of applying ETA; JADAS-27, Juvenile Arthritis Disease Activity Score-27; MTX, Methotrexate; EC, Encorton; SSA, Sulfasalazin; WBC, white blood cell; RBC, red blood cell; Hb, hemoglobin; Ht, hematocrit; PLT, platelet; GPT, glutamic pyruvic transferase; GOT, glutamic oxaloacetic transaminase; Cr, creatinine; CRP, C-reactive protein; ESR, erythrocyte sedimentation rate; RF, rheumatoid factor; N, negative.

**Table 2 biomolecules-13-01737-t002:** The distribution patterns of urine sGAGs, CS/DS, HS, and HA in the control subjects and JIA patients.

		Control Subjects	JIA Patients			
		C *n* = 30 (F/M = 20/10)	TP *n* = 59 (F/M = 40/19)	TR *n* = 24 (F/M = 15/9)	T0 *n* = 35 (F/M = 25/10)	T24 *n* = 35 (F/M = 25/10)
**sGAGs** [mg/g creatinine]	all	8.71 ± 1.86	7.34 ± 2.20 (*p* = 0.023)	6.39 ± 2.55 (*p* = 0.001)	7.02 ± 1.92 (*p* = 0.009)	7.00 ± 2.59 (*p* = 0.009)
	F	9.31 ± 1.71	7.57 ± 1.95 (*p* = 0.011)	6.54 ± 2.81 (*p* = 0.001)	7.13 ± 1.92 (*p* = 0.003)	6.68 ± 2.35 (*p* = 0.000)
	M	7.52 ± 1.61	6.85 ± 2.65 (NS)	6.16 ± 2.17 (*p* = 0.042)	6.73 ± 2.00 (NS)	7.81 ± 3.09 (NS)
**CS/DS** [mg/g creatinine]	all	5.40 ± 1.51	4.17 ± 2.01 (*p* = 0.012)	4.70 ± 1.67 (NS)	3.91 ± 1.90 (*p* = 0.005)	4.66 ± 1.93 (NS)
	F	5.84 ± 1.39	4.30 ± 1.83 (*p* = 0.008)	4.82 ± 1.91 (*p* = 0.049)	3.93 ± 2.00 (*p* = 0.002)	4.41 ± 1.79 (*p* = 0.032)
	M	4.52 ± 1.40	3.88 ± 1.37 (*p* = 0.039)	4.50 ± 1.22 (NS)	3.87 ± 1.75 (NS)	5.28 ± 2.21 (NS)
**HS** [mg/g creatinine]	all	1.29 (0.90; 1.49)	2.74 (1.72; 4.36) (*p* = 0.000)	1.08 (0.71; 1.70) (NS)	2.99 (1.55; 4.11) (*p* = 0.000)	1.10 (0.70; 1.51) (NS)
	F	1.32 (0.90; 1.54)	3.03 (1.95; 4.37) (*p* = 0.000)	0.95 (0.64; 2.02) (NS)	3.41 (1.71; 4.41) (*p* = 0.002)	1.10 (0.69; 1.51) (NS)
	M	1.08 (0.83; 1.49)	2.38 (1.64; 4.26) (*p* = 0.020)	1.22 (0.78; 1.64) (NS)	2.80 (1.35; 3.94) (NS)	1.09 (0.86; 1.37) (NS)
**HA** [µg/g creatinine]	all	20.25 ± 12.36	31.70 ± 15.55 (*p* = 0.000)	21.24 ± 10.07 (NS)	31.49 ± 14.95 (*p* = 0.002)	15.25 ± 7.04 (NS)
	F	20.0 ± 12.15	32.24 ± 16.28 (*p* = 0.006)	20.83 ± 12.24 (NS)	31.19 ± 15.21 (*p* = 0.030)	15.72 ± 7.55 (NS)
	M	20.57 ± 8.30	30.56 ± 13.85 (NS)	21.91 ± 5.34 (NS)	32.25 ± 15.04 (*p* = 0.042)	16.56 ± 5.69 (NS)

Results are expressed as mean ± SD or medians (quartile 1 and quartile 3); JIA, juvenile idiopathic arthritis; *p* < 0.05 compared to control group; NS, not statistically significant; C, control subjects; TP, patients with newly diagnosed and previously untreated JIA; TR, JIA patients who achieved remission with MTX, EC, and SSA; T0, JIA patients with aggressive disease before the start of ETA treatment; T24, JIA patients after twenty-four months of biological treatment; F, female; M, male; sGAGs, sulfated glycosaminoglycan; CS/DS, chondroitin sulfate/dermatan sulfate glycans; HS, heparan sulfate; HA, hyaluronic acid.

**Table 3 biomolecules-13-01737-t003:** Correlation analysis between urine sGAGs, CS/DS, HS, HA, and serum CRP levels in control subjects and JIA patients.

Parameter	Control Subjects	JIA Patients			
	C *n* = 30	TP *n* = 59	TR *n* = 24	T0 *n* = 35	T24 *n* = 35
sGAGs r(p)	0.5108 (*p* = 0.004)	−0.0620 (NS)	0.7742 (*p* = 0.000)	−0.0707 (NS)	0.7377 (*p* = 0.000)
CS/DS r(p)	0.6417 (*p* = 0.000)	−0.5821 (*p* = 0.000)	0.7821 (*p* = 0.000)	−0.6508 (*p* = 0.000)	0.7611 (*p* = 0.000)
HS r(p)	−0.4256 (*p* = 0.019)	0.7221 (*p* = 0.000)	0.7104 (*p* = 0.000)	0.8048 (*p* = 0.000)	0.2956 (NS)
HA r(p)	0.3344 (NS)	0.6902 (*p* = 0.000)	0.6389 (*p* = 0.001)	0.6360 (*p* = 0.000)	0.5186 (*p* = 0.001)

Results are expressed as Pearson’s correlation coefficients (r); JIA, juvenile idiopathic arthritis; C, control subjects; TP, patients with newly diagnosed and previously untreated JIA; TR, JIA patients who achieved remission with MTX, EC, and SSA; T0, JIA patients with aggressive disease before the start of ETA treatment; T24, JIA patients after twenty-four months of biological treatment; sGAGs, sulfated glycosaminoglycan; CS/DS, chondroitin sulfate/dermatan sulfate glycans; HS, heparan sulfate; HA, hyaluronic acid; CRP, C-reactive protein; NS, not statistically significant.

## Data Availability

The datasets analyzed or generated during the study are available from the authors (winsz@sum.edu.pl; kkuznik@sum.edu.pl).

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
