# Peer review of "Investigation of Glycosaminoglycans in Urine and Their Alteration in Patients with Juvenile Idiopathic Arthritis"

_biomolecules, 2023, doi:10.3390/biom13121737_

Round 1
Reviewer 1 Report
Comments and Suggestions for Authors
Thank you for the invitation to review an interesting article entitled "Investigation of Glycosaminoglycans in Urine and their Alteration in Patients with Juvenile Idiopathic Arthritis." The title and abstract correspond well with the content of the article. The article is written clearly and concisely. The introduction section is sufficient for the purpose, as well as Material and methods. Results are clearly presented.
Statistical analysis is correct.
However, in Table 1 and Table 2 p-values should be presented inside the Table on the right to facilitate the reading of the results. Accordingly, they are not needed in the text.
The article does not contain a conclusion section. Otherwise, the discussion is clear. References are correct.
The conclusion should be placed in the abstract section.
Comments on the Quality of English LanguageMinor spell check required
Author Response
We would like to thank the Reviewer for the evaluation of our article. We are grateful for the important comments, which we fully addressed in the revised manuscript. The text of our manuscript (in each individual part, changes made in blue) has been modified, so as to facilitate its understanding and make it acceptable for publication.
Detailed modifications are presented below:
- Following the Reviewer’s recommendation, we have modified the part of the "Results" section. We have modified Tables: 1 [p.4, lines 133] and 2 [p. 7, lines 249]. The p-values are shown inside Table 2, on the right-hand side [p. 7, lines 249]. Accordingly, we have modified their citation (i.e. p<0.05) in the text [p.6, lines 218-238 and p. 7, lines 239-247].
- Following the Reviewer’s recommendation, we have added a section “Conclusion” [p.13, lines 488-501], and we have also modified the part of the "Abstract" section [p.1, lines 28-34].
An English native speaker has reviewed the manuscript and language mistakes have been corrected.
We would like to thank you very much for the evaluation of our paper.
Reviewer 2 Report
Comments and Suggestions for Authors
The paper by Lato-Kariakin and cols. analyses glycosaminoglycans in urine of JIA patients, with an emphasis on patients treated with etanercept, which can be of interest to the field. Some points about the sample collection timeline need clarification.
Specific points:
Table 1 – please provide a better delineation of the columns under ‘during treatment of ETA’ . Also, please replace with ‘During ETA treatment’.
What are the JIA types of these patients?
Line 116 - as normal for this age group – please provide reference – literature, clinical information?
When was the urine from the control group collect? During routine control tests, or from the first morning micturition? What was the time interval between urine collection and processing? And how was the urine from the JIA patients collected?
How long were the patients in the TR group treated? When was the urine from group TR collected?
How long were patients treated before they started on ETA?
Please provide a figure with a timeline showing when the samples from the different groups were collected.
Line 99: After 3 months of effective therapy – please clarify. Are these the TR group?
Comments on the Quality of English LanguageThe paper is overall clearly written but I recommend a revision of English usage, especially word order.
Author Response
We would like to thank the Reviewer for the evaluation of our article. We are grateful for the important comments, which we fully addressed in the revised manuscript. The text of our manuscript (in each individual part) has been modified, so as to facilitate its understanding and make it acceptable for publication.
Detailed modifications are presented below:
- Following the Reviewer’s recommendation, we have modified the part of the "Materials and Methods" section, as follows:
- we have added a detailed description of the columns in Table 1, and we used the term “During ETA treatment” [p.4, lines 133, Table 1],
- we have added information on JIA types [p. 2, lines 89-90],
- we have described the control group in more detail, and we have added literature [p. 3, lines 120-123],
- we have described in detail the principles of collecting urine samples from healthy children and patients with JIA [p. 4, lines 138-146],
- we have completed the description of the treatment of JIA patients from the TR groupand others [p.3, lines 98-110],
- we have added a figure with a timeline showing when the samples from the different groups were collected [p. 5, lines 147-153].
An English native speaker has reviewed the manuscript and language mistakes have been corrected.
We would like to thank you very much for the evaluation of our paper.